# Prevalence and associated factors of dyslipidemia among adults with coexisting chronic disease in Ethiopia: A systematic review and meta-analysis

Tadesse Mekonen Zeleke[1]*, Fitsum Endale Liben[2], Tinsae Shemelise Tesfaye[3], Abiyu Legesse Munea[4], Fekede Asefa Kumsa[5,6]

1 Food Science and Nutrition Research Directorate, Ethiopian Public Health Institute, Addis Ababa, Ethiopia, 2 Department of Public Health, College of Medicine and Health Sciences, Wachemo University, Hossana, Ethiopia, 3 Institute of Physical Activity and Nutrition (IPAN), School of Exercise and Nutrition Sciences, Deakin University, Melbourne, Victoria, Australia, 4 School of Medicine and Public Health, College of Medicine and Health Sciences, Wolaita Sodo University, Wolaita Sodo, Ethiopia, 5 Center for Biomedical Informatics, Department of Pediatrics, College of Medicine, University of Tennessee Health Science Center-Oak Ridge National Laboratory (UTHSC-ORNL), Memphis, Tennessee, United States of America, 6 School of Public Health, College of Health and Medical Sciences, Haramaya University, Harar, Ethiopia

* anguach257@gmail.com

## Abstract

### Introduction

Dyslipidemia is a well-established, modifiable risk factor for cardiovascular disease. This study aimed to determine the pooled prevalence of dyslipidemia, its components, and associated factors among adult patients residing in Ethiopia.

### Method

A comprehensive search of articles was conducted using MEDLINE, Embase, African Journals Online, Google Scholar, and University repositories. The quality of the studies was evaluated using the Joanna Briggs Institute quality appraisal criteria. Dyslipidemia was defined based on the National Cholesterol Education Program Adult Treatment Panel III cut-off points. We pooled the effect sizes using a random-effects model in Stata (v.16). Subgroup and sensitivity analysis were also performed. Small study effect was assessed by Luis Furuya-Kanamori asymmetry index, Doi plot, funnel plot, and Egger's tests.

### Results

Out of 640 initially identified articles, 104 underwent full-text review, and 62 met the inclusion criteria for the final analysis. The pooled prevalence of overall dyslipidemia was 64% (95% CI: 54–74%; $I^2$ = 98.74; P < 0.001). The prevalence of elevated total cholesterol was 32% (95% CI: 27–36%; $I^2$ = 97.09; P < 0.001), elevated total

**Data availability statement:** All relevant data are within the paper and its Supporting Information files.

**Funding:** The author(s) received no specific funding for this work.

**Competing interests:** The authors have declared that no competing interests exist.

triglycerides was 43% (95% CI: 38–47%; $I^2$ = 97.56; P < 0.001), elevated low-density lipoprotein was 30% (95% CI: 24–36%; $I^2$ = 97.24; P < 0.001), and low high-density lipoprotein was 48% (95% CI: 42–54%; $I^2$ = 98.53; P < 0.001). Having high wrist circumference, older age, being overweight or obese, residing in urban, being an alcohol consumer, being a smoker, and being physically inactive were associated with increased odds of dyslipidemia.

## Conclusion

Dyslipidemia prevalence is alarmingly high among adults with coexisting chronic diseases in Ethiopia, posing a significant public health challenge. Regular screening, early detection, and prompt management of dyslipidemia are critical to addressing this issue.

## Introduction

Dyslipidemia refers to the presence of lipid disorders characterized by either one or any combination of elevated levels of total cholesterol (TC), low-density lipoprotein cholesterol (LDL-c), triglycerides (TG), or low levels of high-density lipoprotein cholesterol (HDL-c) [1]. The global burden of lipid abnormalities has increased over the past 30 years [2]. These abnormalities have been strongly linked to higher mortality rates, increased morbidity, and substantial healthcare expenses [3].

Globally, dyslipidemia remains a largely underdiagnosed and undertreated condition [2]. Studies indicate a high prevalence of dyslipidemia among adults with coexisting chronic diseases such as mental disorders [4], hypertension, diabetes mellitus, metabolic syndrome, and HIV (human immunodeficiency virus) patients [5]. The World Health Organization (WHO) estimated in 2008 that Africa had a lower prevalence of elevated TC 23.1% in adults [6]. However, hospital-based studies in Ethiopia have reported a prevalence of dyslipidemia as high as 93.2% among adults with chronic disease [7–9].

Dyslipidemia is a well-established modifiable risk factor for cardiovascular diseases (CVD) [10,11]. Individuals with dyslipidemia face a twofold higher risk of developing CVD compared to those with normal lipid levels [11]. CVD is a chronic non-communicable disease (NCD) and one of the leading causes of global mortality. According to the American Heart Association, the annual number of CVD-related deaths was 19 million in 2020 and is estimated to reach 22.2 million in 2030 [12,13]. Over the past four decades, there has been a significant surge in the prevalence of cardiovascular diseases (CVD) in Africa. And it is anticipated to overtake infectious diseases as the leading cause of death by 2030 [14]. Ethiopia has made inadequate progress in reducing cardiovascular and other non-communicable diseases, with CVDs are one of the leading causes of premature mortality and death [15,16]. Access to public healthcare facilities and availability of health services in Ethiopia are generally limited.

Furthermore, there are substantial disparities in accessibility and inequalities in the distribution of healthcare resources [17–19]. These healthcare challenges might also contribute to the rising prevalence of dyslipidemia and present significant barriers to its prevention and management.

This review will enable researchers to evaluate the burden of dyslipidemia, where existing evidence is often hospital-based, fragmented and inconsistent. Understanding the prevalence and factors associated with dyslipidemia can also provide valuable insights for researchers and policymakers, in designing targeted interventions and strategies for effective disease management and the prevention of further complications. Therefore, this study aimed to systematically review and pool the prevalence of dyslipidemia, its components, and associated factors among adults with coexisting chronic disease in Ethiopia.

## Methods

The study followed Preferred Reporting Items for Systematic Review and Meta-Analysis (PRISMA-2020) guideline for design and reporting (S1 Table). The review protocol is registered with International Prospective Register of Systematic Reviews (CRD42023438436) and can be accessed online at https://www.crd.york.ac.uk/PROSPERO/#myprospero.

### Search strategy

We systematically searched MEDLINE, Embase, and African Journals Online for articles published in Englishchildren between January 1, 2014, and October 23, 2023. Reference lists of included studies were also reviewed for additional studies. Furthermore, Grey literature was identified through Google Scholar, and University repositories. The search strategy included: "dyslipid*", "dyslipidemia", "dyslipidaemia", "lipid profile", "metabolic syndrome*" "Hypercholesterol*", "hyperlipidemia", "lipid disorder", "Prevalence", "associated factors", and "Ethiopia". A detailed list of search terms is provided in Supplementary Information (S2 Table).

### Eligibility criteria

We included original studies reporting the prevalence and factors associated with overall dyslipidemia, elevated TC, LDL-c, TG, or low HDL-c among adult patients (aged ≥ 18 years) in Ethiopia, as no eligible studies on dyslipidemia in children and adolescents with coexisting chronic illnesses were identified. Studies on metabolic syndrome were also included if dyslipidemia or its components were independently identified. Only studies meeting the Adult Treatment Panel III case definition of dyslipidemia were considered. We excluded conference abstracts, editorials, commentaries, reviews, letters, case reports, and studies without primary data. Additionally, studies on the general population and Ethiopians living abroad were excluded.

### Outcome measurement

The primary outcome of this review was dyslipidemia and its components. Definitions of dyslipidemia varied across studies. Therefore, for this review, it was defined as one or more of the following: elevated TC ≥ 200 mg/dl (5.17 mmol/L), elevated LDL-c ≥130 mg/dl (3.36 mmol/L), elevated TG ≥150 mg/dl (1.7 mmol/L), and low HDL-c <40 mg/dl (1.03 mmol/L) or low HDL-c <40 mg/dl (1.03 mmol/L) for men and <50 mg/dl (1.3 mmol/L) for women [20,21].

### Study selection

Studies were identified using search terms and database filters, then imported them into covidence, a systematic review software (Melbourne, Australia, Veritas Health Innovation) to remove duplicates and conduct screening. Two investigators (TMZ and TST) independently screened the titles, abstracts and full texts of potentially eligible studies and assessed their

quality. Discrepancies were resolved through consensus. Articles meeting both eligibility and quality criteria were included in the review.

### Data extraction

Data were extracted using a structured template in Microsoft Excel. Extracted variables included the first authors' name, year of publication, region, study settings, study design, sample size, population-specific diseases, age, frequency of overall dyslipidemia, frequency of dyslipidemia components (elevated TC, LDL-c, TG, and low HDL-c), diagnostic cut-off, and associated factors. To ensure data quality and reliability, two independent reviewers (TMZ and FEL) conducted the extraction process. Any discrepancies were resolved through consensus (S1 File).

### Quality assessments

The quality of the included studies was independently assessed by two researchers (TMZ and TST) using Joanna Briggs Institute's (JBI's) critical appraisal checklist for prevalence studies [22], cohort studies [23], and case control [24] studies. These tools comprised nine, ten, and eleven questions, respectively, with response options of "yes," "no," "unclear," or "not applicable." Based on the raters' evaluations, studies were categorized as "include," "exclude," or "seek further information." Discrepancies between the two reviewers were resolved through discussion (S3 Table).

### Statistical analysis

Data analysis was performed using Stata version 16.0 (Stata Corp LP, College Station, TX, USA). The prevalence of dyslipidemia and its components was pooled using a random-effects model, with pooled prevalence presented in a forest plot. We conducted a heterogeneity assessment using Higgins' I-squared ($I^2$) statistics, performed subgroup analyses, and evaluated small-study effects, including tests for publication bias. Small-study effects were initially assessed through funnel plot asymmetry inspection and Egger's regression p-value. However, the combined studies only partially met the criteria for employing a funnel plot [25]. Consequently, the Luis Furuya-Kanamori asymmetry(LFK) index and Doi plot were used alongside the funnel plot to examine asymmetry and detect small study effects [26]. Sensitivity analysis was conducted to evaluate the influence of individual studies on the overall estimate. Statistical significance was defined as a P-value of less than 0.05.

## Results

### Literature search

The initial literature searches yielded 640 records. Of these, 170 duplicated were excluded. A total of 470 articles were screened based on their titles and abstracts, and 104 were underwent for full text assessment. Finally, 62 articles were included in this systematic review and meta-analysis (Fig 1 and S2 File).

### Characteristics of the included studies

The included 62 studies consist of 18,939 adult patients. These studies consisted of 59 cross-sectional studies (out of which eight studies were comparative cross-sectional studies), two cohort studies, and one case-control study. Publication dates for the included studies range from 2014 to 2023. Most of the studies (43 out of 62) were conducted in one city (Addis Ababa) and two Ethiopian regional (Amhara and Oromia) administrations. These studies examined the prevalence of overall dyslipidemia or dyslipidemia components (elevated TC, TG, LDL-c, or low HDL-c). Seventeen studies examined dyslipidemia among 6,076 HIV patients, while another 17 studies investigated dyslipidemia in 5431 diabetic patients. Table 1 presents the basic characteristics of the included studies.

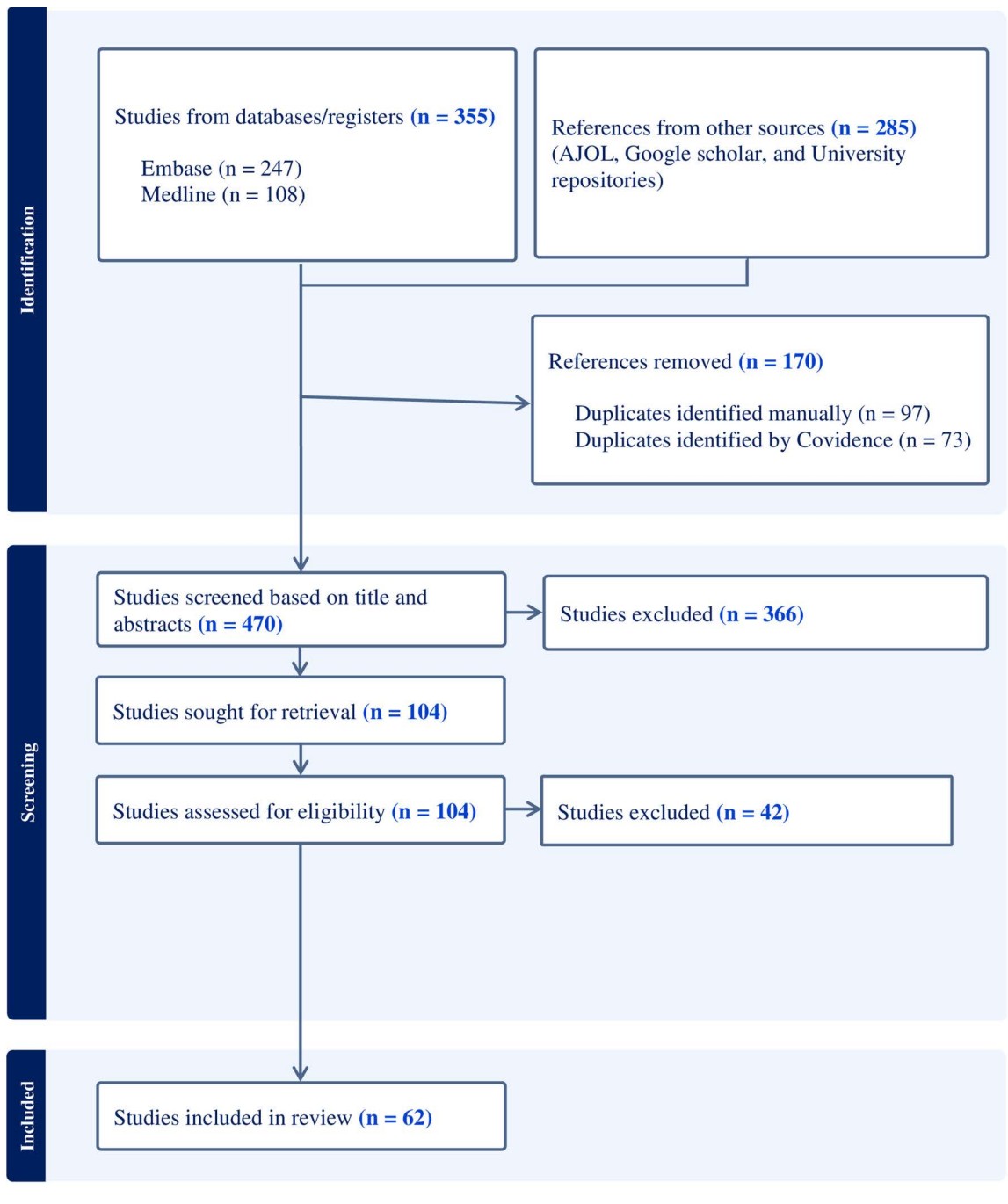

**Fig 1. PRISMA 2020 flow diagram depicting the selection process.**

**Table 1.** Summary characteristics of studies included in the meta-analysis.

| Author and publication year | Region | Study design | Study population | Sample size | Prevalence of overall dyslipidemia (%) | Prevalence of elevated TC (%) | Prevalence of elevated TG (%) | Prevalence of elevated LDL-c (%) | Prevalence of low HDL-c (%) | Factors associated with dyslipidemia |
|---|---|---|---|---|---|---|---|---|---|---|
| Addisu et al. 2023 | Oromia | Cross-sectional | Cardiac Patients | 269 | 76.6 | 39.8 | 44.6 | 29.4 | 53.5 * | Gender, marital status, residence, physical exercise, alcohol consumption, history of alcohol consumption, and smoking |
| Assefa et al. 2023 | Addis Ababa | Cross-sectional | HIV patients | 288 | 55.2 | 22.6 | 18.8 | 4.9 | 48.6 ** | |
| Fentie and Yibabie, 2023 | Dire Dawa | Comparative cross-sectional | Psychiatric patients | 96 | – | 42.7 | 38.5 | – | 35.4* | Age, gender, educational status, residence, physical exercise, alcohol consumption, smoking, and BMI |
| Kelem et al. 2023 | Amhara | Cross-sectional | Metabolic syndrome patients | 384 | – | – | 59.4 | – | – | |
| Mohammed et al. 2023 | Amhara | Cross-sectional | Hypertensive patients | 384 | 93.2 | 47.7 | 50.3 | 44.3 | 59.6 * | Gender |
| Abdissa and Hirpa 2022 | Oromia | Cross-sectional | Diabetic patients | 390 | – | 29.2 | 63.3 | – | 47.9 * | Age, gender, marital status, educational status, residence, alcohol consumption, smoking, hypertension, BMI, family history of diabetes mellitus, family history of hypertension, and duration of diabetes mellitus |
| Gebreyesus et al. 2022 | Tigray | Cross-sectional | Diabetic patients | 421 | – | 35.2 | 41.1 | – | 74.6 ** | |
| Hashim et al. 2022 | Addis Ababa | Cross-sectional | Dyspeptic patients | 346 | 73.1 | 23.1 | 20.8 | 58.1 | 55.2 * | Age, gender, alcohol consumption, smoking, helicobacter pylori status, HC, WC, and BMI |
| Kabtu and Tsegaw 2022 | Addis Ababa | Cross-sectional | Diabetic patients | 165 | – | 31.1 | 43.8 | 16 | – | |
| Kassaw et al. 2022 | Amhara | Comparative cross-sectional | Epileptic patients | 102 | – | – | 33.3 | – | 40.2** | |
| Letta et al. 2022 | Harar | Cross-sectional | Diabetic patients | 879 | – | 41.2 | 63.1 | – | – | |
| Lissane et al. 2022 | Addis Ababa | Cross-sectional | Stroke patients | 192 | – | 21.1 | – | 31.5 | – | |
| Nigatie et al. 2022 | Amhara | Comparative cross-sectional | H. pylori-infected patients | 117 | 71.8 | 10.3 | 10.3 | 35 | 45.3** | |

*(Continued)*

**Table 1.** (Continued)

| Author and publication year | Region | Study design | Study population | Sample size | Prevalence of overall dyslipidemia (%) | Prevalence of elevated TC (%) | Prevalence of elevated TG (%) | Prevalence of elevated LDL-c (%) | Prevalence of low HDL-c (%) | Factors associated with dyslipidemia |
|---|---|---|---|---|---|---|---|---|---|---|
| Sahiledengle et al. 2022 | Oromia | Cross-sectional | Diabetic patients | 256 | – | 20.7 | 84.4 | 9.4 | 31.6 ** | |
| Timerga et al. 2022 | SNNPR | Cross-sectional | Metabolic syndrome patients | 324 | – | – | 45.1 | – | 52.2 * | |
| Woldeyes et al. 2022 | Addis Ababa | Cross-sectional | HIV patients | 280 | 67.1 | – | – | – | – | |
| Woldeyes et al. 2022 | Addis Ababa | Cross-sectional | HIV patients | 333 | 69.4 | 49.8 | 36 | 42 | 10.2 * | |
| Woldu et al. 2022 | Addis Ababa | Comparative cross-sectional | HIV patients | 510 | – | – | – | – | 38** | |
| Amogne et al. 2021 | Addis Ababa | Cross-sectional | Psychiatric patients | 510 | – | – | 24.7 | – | 34.5 ** | |
| Bogale and Aderaw 2021 | Amhara | Cross-sectional | Hypertensive patients | 203 | – | 50 | 51.9 | – | 72.4* | |
| Challa et al. 2021 | Addis Ababa | Cross-sectional | Psychiatric patients | 200 | – | – | 30.5 | – | 44 ** | |
| Fanta et al. 2021 | Addis Ababa | Prospective cohort | Acute Coronary Syndrome patients | 176 | 51.1 | – | 31.3 | – | 37.5** | |
| Fiseha et al. 2021 | Amhara | Cross-sectional | HIV patients | 392 | 59.9 | 47.2 | 30.9 | 29.6 | 17.3 * | |
| Gebremedhin et al. 2021 | Oromia | Case-control | Hypertensive patients | 136 | – | 58.1 | 53.7 | – | 39* | |
| Haile et al. 2021 | Oromia | Cross-sectional | Metabolic syndrome patients | 381 | – | 14.2 | 44.6 | – | 67.2 ** | |
| Hirigo et al. 2021 | Sidama | Cross-sectional | Psychiatric patients | 245 | 58.4 | 24.9 | 26.9 | 19.2 | 69.4 * | Age, gender, marital status, educational status, physical exercise, alcohol consumption, smoking, and BMI |
| Kifl et al. 2021 | Amhara | Cross-sectional | Hypertensive patients | 372 | – | – | – | – | 30.9 * | |
| Worku et al. 2021 | Amhara | Cross-sectional | Diabetic patients | 327 | – | 36.1 | – | – | – | Gender, residence, alcohol consumption, smoking, BMI, and family history of diabetes mellitus |
| Woyesa et al. 2021 | Oromia | Comparative cross-sectional | Diabetic patients | 96 | – | – | – | – | 24** | |

*(Continued)*

Table 1. (Continued)

| Author and publication year | Region | Study design | Study population | Sample size | Prevalence of overall dyslipidemia (%) | Prevalence of elevated TC (%) | Prevalence of elevated TG (%) | Prevalence of elevated LDL-c (%) | Prevalence of low HDL-c (%) | Factors associated with dyslipidemia |
|---|---|---|---|---|---|---|---|---|---|---|
| Abdu et al. 2020 | Oromia | Cross-sectional | H. pylori-infected patients | 369 | 77.5 | 35.2 | 53.1 | 26.8 | 41.7 * | Age, gender, residence, physical exercise, chat chewing, smoking, Helicobacter pylori status, HC, WC, BMI, and family history of hypertension |
| Duguma et al. 2020 | Oromia | Cross-sectional | HIV patients | 271 | 32.5 | 25.8 | 26.2 | 18.5 | 79.3 * | |
| Haile and Timerga 2020 | Oromia | Cross-sectional | Diabetic patients | 248 | – | 13.7 | 48 | – | 50.8 * | Age, gender, marital status, educational status, residence, physical exercise, chat chewing, alcohol consumption, smoking, hypertension, and BMI |
| Kemal et al. 2020 | Addis Ababa | Cross-sectional | HIV patients | 353 | 74.8 | 45.9 | 28.9 | 31.2 | 35.7 ** | Age, sex, marital status, educational status, physical exercise, chat chewing, alcohol consumption, smoking, and BMI |
| Kumie et al. 2020 | Addis Ababa | Comparative cross-sectional | Breast cancer patients | 91 | – | 13.2 | 58.2 | 15.4 | 47.3* | |
| Teshome et al. 2020 | Sidama | Cross-sectional | Psychiatric patients | 245 | – | – | 26.9 | – | 41.2 ** | |
| Timerga et al. 2020 | Oromia | Cross-sectional | Metabolic syndrome patients | 256 | 44.9 | – | – | – | – | |
| Yazie 2020 | Addis Ababa | Prospective cohort | HIV patients | 63 | 73 | 38.1 | 23.8 | 17.5 | 41.3* | |
| Zerga and Bezabih 2020 | Amhara | Cross-sectional | Diabetic patients | 330 | – | – | 51.5 | – | 48.5 ** | |
| Zewudie et al. 2020 | Oromia | Cross-sectional | Stroke patients | 220 | – | 40 | 55.5 | – | 24.5 * | |
| Bahrey et al. 2019 | Tigray | Cross-sectional | Hypertensive patients | 578 | 27.3 | – | – | – | – | |
| Bune et al. 2019 | SNNPR | Comparative cross-sectional | HIV patients | 633 | 59.2 | – | 37 | – | 34.3** | |
| Gebre and Assefa 2019 | SNNPR | Cross-sectional | Diabetic patients | 338 | – | – | – | 49.7 | – | |
| Gebremeskel et al. 2019 | Tigray | Cross-sectional | Diabetic patients | 419 | – | – | 45.1 | – | 34.4 ** | |
| Wube et al. 2019 | SNNPR | Cross-sectional | Diabetic patients | 314 | – | 32.8 | – | – | – | |

(Continued)

Table 1. (Continued)

| Author and publication year | Region | Study design | Study population | Sample size | Prevalence of overall dyslipidemia (%) | Prevalence of elevated TC (%) | Prevalence of elevated TG (%) | Prevalence of elevated LDL-c (%) | Prevalence of low HDL-c (%) | Factors associated with dyslipidemia |
|---|---|---|---|---|---|---|---|---|---|---|
| Adal et al. 2018 | Addis Ababa | Cross-sectional | HIV patients | 594 | – | 16 | 28.6 | – | – | |
| Asaye et al. 2018 | Oromia | Cross-sectional | Psychiatric patients | 360 | – | 3.6 | 25.3 | – | 66.1 ** | |
| Ataro et al. 2018 | Harar | Cross-sectional | HIV patients | 425 | – | 30.1 | – | 23.8 | 42.6 * | |
| Belete et al. 2018 | Amhara | Cross-sectional | Diabetic patients | 159 | – | 51.6 | 62.3 | 57.9 | 31.4 ** | |
| Birarra and Gelayee 2018 | Amhara | Cross-sectional | Diabetic patients | 256 | – | – | 68.8 | – | 67.2 ** | |
| Bosho et al. 2018 | Oromia | Cross-sectional | HIV patients | 268 | – | 25 | 29.9 | – | 49.3 ** | |
| Gebrie et al. 2018 | Amhara | Cross-sectional | Hypertensive patients | 100 | – | 52 | – | 54 | – | |
| Hirigo and Geleta 2018 | Sidama | Cross-sectional | Hypertensive patients | 238 | 90.8 | 38.7 | 62.2 | 21 | 60.9 * | |
| Bekele et al. 2017 | SNNPR | Cross-sectional | Diabetic patients | 224 | – | 23.7 | 40.6 | – | 42 * | Age, sex, educational status, residence, hypertension, BMI, family history of DM, and duration of DM |
| Tadewos et al. 2017 | Sidama | Cross-sectional | Hypertensive patients | 238 | – | – | 62.2 | – | 60.9 ** | |
| Woyesa et al. 2017 | Sidama | Cross-sectional | Diabetic patients | 314 | – | – | 70.4 | – | 39.2 ** | |
| Abebe et al. 2016 | Amhara | Cross-sectional | HIV patients | 462 | – | 18.6 | 28.8 | 17.7 | 64.9 * | |
| Hirigo and Tesfaye 2016 | Sidama | Cross-sectional | HIV patients | 185 | – | 42.2 | 44.9 | 30.8 | 70.3 ** | |
| Ambachew et al. 2015 | Sidama | Cross-sectional | Diabetic patients | 295 | – | 34.6 | 29.8 | 34.9 | 12.2 * | |
| Mohammed et al. 2015 | Oromia | Cross-sectional | HIV patients | 393 | – | 34.6 | 66.2 | 39.9 | 89.8 * | |
| Abebe et al. 2014 | Addis Ababa | Comparative cross-sectional | HIV patients | 252 | – | 26.6 | 38.9 | 15.1 | 61.9* | |
| Tachebele et al. 2014 | Amhara | Cross-sectional | Hypertensive patients | 300 | – | – | 27.3 | – | 81.3 ** | |
| Tesfaye et al. 2014 | Sidama | Cross-sectional | HIV patients | 374 | – | 26.2 | – | – | – | |

*HDL-c cut-off < 40mg/dl and ** HDL-c cut-off < 40mg/dl < 40mg/dl for males and < 50 mg/dl for female; - value not mentioned (either did not mentioned in the original study or used a different cut-off value); HDL-c: High Density Lipoprotein-cholesterol; LDL-c: Low Density Lipoprotein-cholesterol; TG: Triglyceride; BMI: Body Mass Index; HIV: Human immune virus; WC: Waist circumference; HC: Hip circumference.

## Prevalence of overall dyslipidemia

Among Eighteen studies that examined the prevalence of overall dyslipidemia, nine studies [7,27–34] comprising 2672 patients' assessed overall dyslipidemia using the NCEP III criteria (i.e., at least one of the following abnormal lipid profiles: TC ≥ 200 mg/dl, TGs ≥ 150 mg/dl, LDL-c ≥ 130 mg/dl, or HDL-c < 40 mg/dl). In a similar manner, the remaining nine studies [8,35–42] consisting of 2919 patients, also assessed dyslipidemia using NCEP III criteria but further considered a sex difference in HDL-c cut-off (HDL < 40 mg/dl for males and < 50 mg/dl for female). Forty-one studies reporting on elevated TC (≥ 200 mg/dl) [7–9,27–35,37,40,43–69], forty-nine on TG (≥ 150 mg/dl) [7–9,27–35,37,38,40,41,43,45–49,51–56,59,60,62–67,69–81], Twenty six on LDL-c (≥ 130 mg/dl) [7–9,27–35,37,40,44,45,47,55,58,60,61,65–67,69,82] and forty-nine on low HDL-c using either HDL-c cut-offs < 40 mg/dl for both sexes [7–9,27–34,43,49,52–56,61,64,66,67,69,71,83] or < 40 mg/dl for male and < 50 mg/dl for female [35,37,38,40,41,45,48,51,59,60,62,65,72–81,84,85]. Accordingly, the prevalence of overall dyslipidemia ranges from 27.3% [42] to 93.2% [7], with a pooled prevalence of 64% (95% CI: 54–74%; $I^2$ = 98.74; P < 0.001). The prevalence of TC ranges from 3.6% in psychiatric patients [62] to 58.1% [52] in hypertensive patients, with the pooled estimate of 32% (95% CI: 27–36%; $I^2$ = 97.09; P < 0.001). Similarly, the prevalence of TG ranges from 10.3% among helicobacter pylori infected patients [37] to 84.4% in diabetic patients [45], with a pooled prevalence of 43% (95% CI: 38–47%; $I^2$ = 97.56; P < 0.001). Likewise, the prevalence of elevated LDL-c ranges from 4.9% [35] among HIV patients to 58.1% [29] among dyspeptic patients, with a pooled prevalence of 30% (95% CI: 24–36%; $I^2$ = 97.24; P < 0.001). In a similar fashion, the prevalence of low HDL-c ranges from 10.2% [28] to 89.8% [9]. The pooled prevalence of HDL-c was 48% (95% CI: 42–54%; $I^2$ = 98.53; P < 0.001). Regardless of which HDL-c cut-off is used, the prevalence of HDL-c remained the same with HDL-c cut-off value < 40 mg/dl in both sexes 48% (95% CI: 38–58%; $I^2$ = 98.99; P < 0.001) and with HDL-c cut-off value < 40 mg/dl for male and < 50 mg/dl for female, 48% (95% CI: 41–55%; $I^2$ = 97.38; P < 0.001). The heterogeneity between groups was statistically significant (P < 0.001) (Figs 2 and 3).

## Factors associated with overall dyslipidemia

This study showed that patients with higher WC (≥ 94 cm) had 3 times more odds to develop dyslipidemia than patients with WC < 94 cm. With respect to age, the pooled odds ratio from eight studies revealed that patients aged > 40 years or > 30 years have higher odds of dyslipidemia than their respective younger age groups. Our study also identified that patients who were overweight or obese, urban residents, and alcohol consumers had more odds to experience dyslipidemia. Similarly, smokers and patients who were physically inactive were associated with having higher odds of dyslipidemia. In contrast, this meta-analysis reported that gender, educational status, and marital status had no association with dyslipidemia (Table 2).

## Sensitivity and sub-group analyses

Due to heterogeneity among studies, we performed several analyses to identify the source of heterogeneity. Initially, we carried out sensitivity analysis by excluding outliers from overall dyslipidemia [7,8], TC [62], TG [37,45], LDL-c [35,45], and HDL-c [7,28,67] to check their effect on the degree of variability. However, the heterogeneity among studies remained high ($I^2$ ≥ 95). Consequently, we included all the studies in the final meta-analysis model. Following this, we conducted sub-group analyses based on region, HDL-c cut-off, and study population.

Our finding suggested dyslipidemia and its components were highly prevalent across some of regional administrations in Ethiopia. We observed variation in the pooled estimate of overall dyslipidemia ranges from 82% (95% CI: 79–85%) in Sidama to 58% (95% CI: 36–80%; $I^2$ = 98.64; P < 0.001) in Oromia region. Additionally, the pooled estimate of elevated TC ranges from 39% (95% CI: 27–51%; $I^2$ = 97.05; P < 0.001) in Amhara to 28% (95% CI: 19–38%; $I^2$ = 98.15; P < 0.001) in Oromia region. Furthermore, the pooled estimate of elevated TG ranges from 61% (95% CI: 58–64%) in Harari and Dire Dawa to 31% (95% CI: 27–36%; $I^2$ = 88.58; P < 0.001) in Addis Ababa. Moreover, the pooled estimate of elevated LDL-c ranges from 39% (95% CI: 27–52%; $I^2$ = 96.72; P < 0.001) in Amhara to 25%

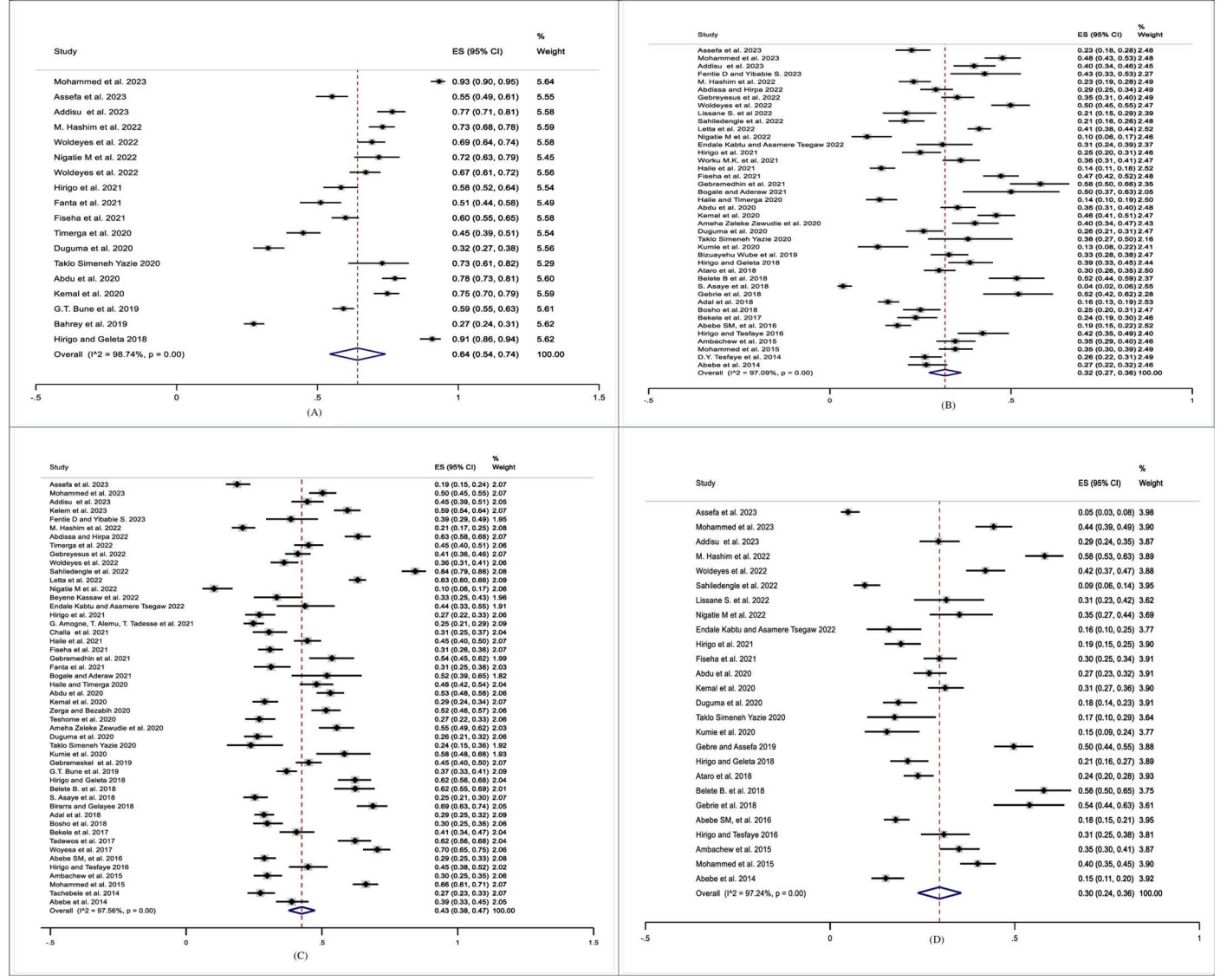

**Fig 2. Forest plots displaying pooled prevalence of overall dyslipidemia and its components among patients in Ethiopia.** A, Forest plot displaying the pooled prevalence of overall dyslipidemia. B, Forest plot displaying the pooled prevalence of TC. C, Forest plot displaying the pooled prevalence of TG. D, Forest plot displaying the pooled prevalence of LDL-c.

(95% CI: 14–36%; I² = 96.50; P < 0.001) in Oromia. Regarding HDL-c, the pooled prevalence ranged from 56% (95% CI: 53–59%) in Tigray to 41% (95% CI: 31–51%; I² = 97.47; P < 0.001) in Addis Ababa. The degree of heterogeneity between groups in each subgroup analysis was statistically significant (S1 Fig). Our sub-group analysis also suggested that a higher prevalence of overall dyslipidemia, 68% (95% CI: 56–81%; I² = 98.39; P < 0.001), was seen among studies with HDL-c cut-off value < 40 mg/dl.

Moreover, there was a significant variation in the pooled prevalence of dyslipidemia and its components among the various study populations. The highest pooled prevalence of elevated TC has been observed among hypertensive patients, 49% (95% CI: 42–56%; I² = 73.42; P < 0.001), while the least was seen among psychiatric patients, 23% (95%

| Study | ES (95% CI) | % Weight |
|---|---|---|
| **HDL < 40 mg/dl** | | |
| Mohammed et al. 2023 | 0.60 (0.55, 0.64) | 2.06 |
| Addisu et al. 2023 | 0.54 (0.48, 0.59) | 2.05 |
| M. Hashim et al. 2022 | 0.55 (0.50, 0.60) | 2.06 |
| Abdissa and Hirpa 2022 | 0.48 (0.43, 0.53) | 2.06 |
| Hirigo et al. 2021 | 0.69 (0.63, 0.75) | 2.05 |
| Z.D. Kifl et al. 2021 | 0.31 (0.26, 0.36) | 2.06 |
| Haile and Timerga 2020 | 0.51 (0.45, 0.57) | 2.04 |
| Abdu et al. 2020 | 0.42 (0.37, 0.47) | 2.06 |
| Bekele et al. 2017 | 0.42 (0.36, 0.49) | 2.04 |
| Ambachew et al. 2015 | 0.12 (0.09, 0.16) | 2.07 |
| Timerga et al. 2022 | 0.52 (0.47, 0.58) | 2.05 |
| Ataro et al. 2018 | 0.43 (0.38, 0.47) | 2.06 |
| Mohammed et al. 2015 | 0.90 (0.86, 0.92) | 2.08 |
| Abebe SM, et al. 2016 | 0.65 (0.60, 0.69) | 2.07 |
| Ameha Zeleke Zewudie et al. 2020 | 0.25 (0.19, 0.31) | 2.05 |
| Hirigo and Geleta 2018 | 0.61 (0.55, 0.67) | 2.04 |
| Woldeyes et al. 2022 | 0.10 (0.07, 0.14) | 2.08 |
| Fiseha et al. 2021 | 0.17 (0.14, 0.21) | 2.07 |
| Duguma et al. 2020 | 0.79 (0.74, 0.84) | 2.06 |
| Fentie D and Yibabie S. 2023 | 0.35 (0.27, 0.45) | 1.99 |
| Gebremedhin et al. 2021 | 0.39 (0.31, 0.47) | 2.01 |
| Bogale and Aderaw 2021 | 0.72 (0.54, 0.85) | 1.82 |
| Kumie et al. 2020 | 0.47 (0.37, 0.57) | 1.97 |
| Taklo Simeneh Yazie 2020 | 0.41 (0.30, 0.54) | 1.93 |
| Abebe et al. 2014 | 0.62 (0.56, 0.68) | 2.05 |
| Subtotal (I^2 = 98.99%, p = 0.00) | 0.48 (0.38, 0.58) | 50.89 |
| | | |
| **HDL < 40 mg/dl for male and HDL < 50mg/dl for female** | | |
| Assefa et al. 2023 | 0.49 (0.43, 0.54) | 2.05 |
| Kemal et al. 2020 | 0.36 (0.31, 0.41) | 2.06 |
| Tadewos et al. 2017 | 0.61 (0.55, 0.67) | 2.04 |
| Gebreyesus et al. 2022 | 0.75 (0.70, 0.79) | 2.07 |
| Zerga and Bezabih 2020 | 0.48 (0.43, 0.54) | 2.05 |
| Belete B. et al. 2018 | 0.31 (0.25, 0.39) | 2.03 |
| Teshome et al. 2020 | 0.41 (0.35, 0.47) | 2.04 |
| S. Asaye et al. 2018 | 0.66 (0.61, 0.71) | 2.06 |
| G. Amogne, T. Alemu, T. Tadesse et al. 2021 | 0.35 (0.31, 0.39) | 2.07 |
| Gebremeskel et al. 2019 | 0.34 (0.30, 0.39) | 2.06 |
| Woyesa et al. 2017 | 0.39 (0.34, 0.45) | 2.05 |
| Challa et al. 2021 | 0.44 (0.37, 0.51) | 2.03 |
| Birarra and Gelayee 2018 | 0.67 (0.61, 0.73) | 2.05 |
| Tachebele et al. 2014 | 0.81 (0.77, 0.85) | 2.07 |
| Haile et al. 2021 | 0.67 (0.62, 0.72) | 2.06 |
| Hirigo and Tesfaye 2016 | 0.70 (0.63, 0.76) | 2.04 |
| Bosho et al.2018 | 0.49 (0.43, 0.55) | 2.05 |
| Sahiledengle et al. 2022 | 0.32 (0.26, 0.38) | 2.05 |
| Nigatie M et al. 2022 | 0.45 (0.37, 0.54) | 2.00 |
| Woyesa S et al. 2021 | 0.24 (0.17, 0.33) | 2.01 |
| Beyene Kassaw et al. 2022 | 0.40 (0.31, 0.50) | 1.99 |
| G.T. Bune et al. 2019 | 0.34 (0.31, 0.38) | 2.07 |
| Fanta et al. 2021 | 0.38 (0.31, 0.45) | 2.03 |
| Woldu et al. 2022 | 0.38 (0.34, 0.42) | 2.07 |
| Subtotal (I^2 = 97.38%, p = 0.00) | 0.48 (0.41, 0.55) | 49.11 |
| | | |
| Heterogeneity between groups: p = 0.970 | | |
| Overall (I^2 = 98.53%, p = 0.00); | 0.48 (0.42, 0.54) | 100.00 |

**Fig 3. Forest plots displaying pooled prevalence of HDL-c among patients in Ethiopia.**

**Table 2. Summary of the factors associated with overall dyslipidemia among patients in Ethiopia.**

| Characteristic | Exposed group | Comparison group | Study | OR (95% CI) | Pooled OR (95% CI) | I² |
|---|---|---|---|---|---|---|
| Socio-demographic characteristics | | | | | | |
| Age | ≥30 | < 30 | Abdissa and Hirpa, 2022 | 2.71 (1.51-4.85) | 2.13 (0.99–4.62) | 77.5% |
| | | | Haile and Timerga, 2020 | 3.75 (1.78 -7.91) | | |
| | | | Bekele et al. 2017 | 1.02 (0.57 - 1.83) | | |
| Age | > 40 | ≤ 40 | Fentie and Yibabie, 2023 | 1.48 (0.57–3.84) | 2.12 (1.39–3.25) | 61.5% |
| | | | Hashim et al. 2022 | 4.56 (2.65–7.87) | | |
| | | | Hirigo et al. 2021 | 1.44 (0.81–2.58) | | |
| | | | Kemal et al. 2020 | 2.17 (1.30–3.61) | | |
| | | | Abdu et al. 2020 | 1.74 (1.05–2.88) | | |
| Gender | Female | Male | Mohammed et al. 2023 | 0.64 (0.29–1.43) | 1.1 (0.83–1.46) | 64% |
| | | | Addisu et al. 2023 | 0.93 (0.53–1.65) | | |
| | | | Fentie and Yibabie, 2023 | 1.43 (0.55–3.70) | | |
| | | | Hashim et al. 2022 | 0.82 (0.50–1.36) | | |
| | | | Abdissa and Hirpa, 2022 | 2.24 (1.31–3.83) | | |
| | | | Hirigo et al. 2021 | 1.11 (0.66–1.85) | | |
| | | | Worku et al. 2021 | 1.47 (0.94–2.29) | | |
| | | | Haile and Timerga, 2020 | 1.83 (1.07–3.15) | | |
| | | | Abdu et al. 2020 | 0.55 (0.33–0.91) | | |
| | | | Kemal et al. 2020 | 1.56 (0.95–2.57) | | |
| | | | Bekele et al. 2017 | 0.66 (0.38–1.15) | | |
| Marital status | Married | Single, divorced or widowed | Addisu et al. 2023 | 0.60 (0.33–1.10) | 1.03 (0.76–1.41) | 35.2% |
| | | | Abdissa and Hirpa, 2022 | 1.09 (0.67–1.77) | | |
| | | | Hirigo et al. 2021 | 1.55 (0.92–2.62) | | |
| | | | Haile and Timerga, 2020 | 0.80 (0.39–1.65) | | |
| | | | Kemal et al. 2020 | 1.20 (0.71–2.01) | | |
| Educational status | Illiterate | Educated (primary and above) | Fentie and Yibabie, 2023 | 0.36 (0.15–0.84) | 0.94 (0.57–1.55) | 64.4% |
| | | | Abdissa and Hirpa, 2022 | 0.67 (0.37–1.20) | | |
| | | | Hirigo et al. 2021 | 0.49 (0.15–1.60) | | |
| | | | Haile and Timerga, 2020 | 1.38 (0.81–2.36) | | |
| | | | Kemal et al. 2020 | 1.63 (0.76–3.51) | | |
| | | | Bekele et al. 2017 | 1.76 (0.85–3.61) | | |
| Residence | Urban | Rural | Addisu et al. 2023 | 1.43 (0.81–2.53) | 1.44 (0.92–2.26) | 76.1% |
| | | | Fentie and Yibabie, 2023 | 13.43 (3.71–48.64) | | |
| | | | Abdissa and Hirpa, 2022 | 0.99 (0.58–1.68) | | |
| | | | Worku et al. 2021 | 1.48 (0.92–2.38) | | |
| | | | Haile and Timerga, 2020 | 1.26 (0.74–2.15) | | |
| | | | Abdu et al. 2020 | 1.99 (1.21–3.26) | | |
| | | | Bekele et al. 2017 | 0.56 (0.31–0.99) | | |
| Behavioral characteristics | | | | | | |
| Physical activity | No | Yes | Addisu et al. 2023 | 10.16 (3.55–29.02) | 1.28 (0.99–1.66) | 85.8% |
| | | | Fentie and Yibabie, 2023 | 0.28 (0.12–0.67) | | |
| | | | Hirigo et al. 2021 | 0.85 (0.40–1.79) | | |
| | | | Abdu et al. 2020 | 0.53 (0.24–1.17) | | |
| | | | Haile and Timerga, 2020 | 1.96 (1.14–3.38) | | |
| | | | Kemal et al. 2020 | 1.11 (0.66–1.88) | | |

*(Continued)*

 

| Characteristic | Exposed group | Comparison group | Study | OR (95% CI) | Pooled OR (95% CI) | I² |
|---|---|---|---|---|---|---|
| Chat chewing | Yes | No | Abdu et al. 2020 | 1.47 (0.85–2.53) | 1.13 (0.70–1.82) | 26.2% |
| | | | Haile and Timerga, 2020 | 0.79 (0.44–1.39) | | |
| | | | Kemal et al. 2020 | 1.71 (0.37–7.97) | | |
| Current alcohol consumption | Yes | No | Addisu et al. 2023 | 2.76 (0.81–9.48) | 1.46 (0.99–2.16) | 45.9% |
| | | | Abdissa and Hirpa, 2022 | 2.48 (1.32–4.63) | | |
| | | | Hashim et al. 2022 | 1.78 (1.06–2.97) | | |
| | | | Hirigo et al. 2021 | 0.70 (0.24–2.06) | | |
| | | | Haile and Timerga, 2020 | 1.30 (0.63–2.69) | | |
| | | | Kemal et al. 2020 | 0.94 (0.57–1.54) | | |
| History of alcohol consumption | Yes | No | Addisu et al. 2023 | 2.70 (1.16–6.30) | 2.21 (1.20–4.07) | 44.2% |
| | | | Fentie and Yibabie, 2023 | 4.37 (1.36–14.02) | | |
| | | | Worku et al. 2021 | 1.48 (0.90–2.43) | | |
| Currently Smoking | Yes | No | Addisu et al. 2023 | 1.57 (0.44–5.61) | 1.26 (0.84–1.90) | 0.0% |
| | | | Abdissa and Hirpa, 2022 | 1.34 (0.57–3.12) | | |
| | | | Hashim et al. 2022 | 0.55 (0.07–4.62) | | |
| | | | Hirigo et al. 2021 | 1.47 (0.53–4.04) | | |
| | | | Abdu et al. 2020 | 2.22 (0.65–7.62) | | |
| | | | Haile and Timerga, 2020 | 0.81 (0.34–1.93) | | |
| | | | Kemal et al. 2020 | 1.13 (0.30–4.20) | | |
| Previous history of smoking | Yes | No | Addisu et al. 2023 | 0.97 (0.46–2.04) | 2.05 (0.85–4.90) | 67.7% |
| | | | Fentie and Yibabie, 2023 | 3.75 (0.79–17.81) | | |
| | | | Worku et al. 2021 | 3.00 (1.70–5.31) | | |
| Clinical and anthropometric characteristics | | | | | | |
| Helicobacter pylori | Positive | Negative | Hashim et al. 2022 | 2.19 (1.30–3.68) | 0.84 (0.13–5.51) | 96% |
| | | | Abdu et al. 2020 | 0.32 (0.19–0.55) | | |
| Hypertension | ≥ 140/90 mmHg | < 140/90 mmHg | Abdissa and Hirpa, 2022 | 1.27 (0.75–2.14) | 1.60 (0.97–2.66) | 58.9% |
| | | | Haile and Timerga, 2020 | 2.71 (1.53–4.80) | | |
| | | | Bekele et al. 2017 | 1.21 (0.67–2.20) | | |
| HC | ≥ 102 cm | < 102 cm | Hashim et al. 2022 | 1.93 (1.04–3.59) | 1.92 (1.13–3.25) | 0.0% |
| | | | Abdu et al. 2020 | 1.90 (0.71–5.04) | | |
| WC | ≥ 94 cm | < 94 cm | Hashim et al. 2022 | 1.59 (0.78–3.22) | 3.24 (0.53–19.81) | 67.7% |
| | | | Abdu et al. 2020 | 10.70 (1.44–79.42) | | |
| BMI | ≥ 25 kg/m2 | < 25 kg/m2 | Fentie and Yibabie, 2023 | 2.58 (1.07–6.25) | 1.52 (0.91–2.56) | 82.4% |
| | | | Abdissa and Hirpa, 2022 | 1.22 (0.72–2.06) | | |
| | | | Hashim et al. 2022 | 2.45 (1.40–4.26) | | |
| | | | Hirigo et al. 2021 | 2.16 (1.23–3.79) | | |
| | | | Worku et al. 2021 | 1.59 (0.97–2.62) | | |
| | | | Abdu et al. 2020 | 9.21 (2.20–38.66) | | |
| | | | Haile and Timerga, 2020 | 1.51 (0.85–2.68) | | |
| | | | Kemal et al. 2020 | 1.84 (1.04–3.25) | | |
| | | | Bekele et al. 2017 | 0.07 (0.02–0.20) | | |
| Family history of DM | Yes | No | Abdissa and Hirpa, 2022 | 1.50 (0.81–2.79) | 0.94 (0.57–1.56) | 51.9% |
| | | | Worku et al. 2021 | 0.66 (0.41–1.07) | | |
| | | | Bekele et al. 2017 | 0.91 (0.43–1.92) | | |

*(Continued)*

**Table 2.** (Continued)

| Characteristic | Exposed group | Comparison group | Study | OR (95% CI) | Pooled OR (95% CI) | I² |
|---|---|---|---|---|---|---|
| Family history of hypertension | Yes | No | Abdissa and Hirpa, 2022 | 0.85 (0.46–1.56) | 1.12 (0.60–2.10) | 41.0% |
| | | | Abdu et al. 2020 | 1.62 (0.76–3.46) | | |
| Duration of DM | ≥ 10 years | < 10 years | Abdissa and Hirpa, 2022 | 2.18 (1.00–4.76) | 0.82 (0.12–5.58) | 91.2% |
| | | | Bekele et al. 2017 | 0.31 (0.14–0.70) | | |

OR: odd ratio; CI confidence interval; I²: percentage of variance; BMI: Body Mass Index; WC: Waist circumference; HC: Hip circumference; DM: Diabetes mellitus.

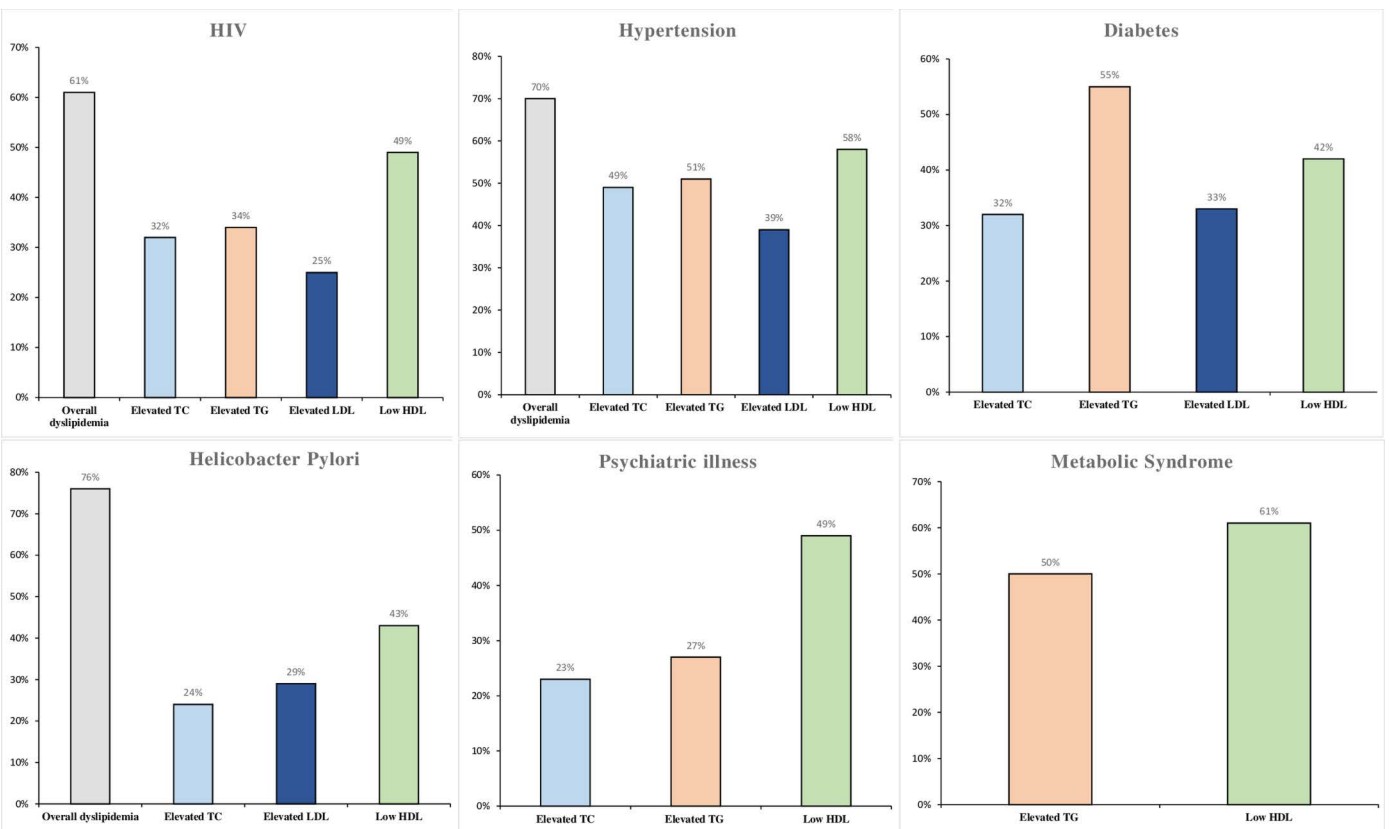

**Fig 4. Pooled prevalence of dyslipidemia and dyslipidemia components in adult patients with HIV, Hypertension, Diabetes, H. pylori positive, Psychiatric illness, and metabolic syndrome.**

CI: 2–44%). The pooled prevalence of elevated TG ranged from 55% (95% CI: 46–63%; I² = 97.19; P < 0.001) among diabetic patients to 27% (95% CI: 24–30%; I² = 41.37; P = 0.13) among psychiatric patients. The pooled prevalence of elevated LDL-c ranged from 39% (95% CI: 21–58%) among hypertensive patients to 25% (95% CI: 17–32%; I² = 96.99; P < 0.001) among HIV patients. The pooled prevalence of low HDL-c ranged from 61% (95% CI: 57–64%) among patients with metabolic syndrome to 42% (95% CI: 30–54%; I² = 98.23; P < 0.001) among diabetic patients. The heterogeneities between groups were statistically significant. Fig 4 and S2 Fig present findings of the subgroup analyses by study population, and HDL-c cut-off.

Fig 4 depicts the prevalence of overall dyslipidemia, elevated total cholesterol, elevated triglyceride, low HDL-c, and elevated LDL-c in patients with HIV, hypertension, diabetes, H. pylori, psychiatric illness and those with metabolic syndrome. The prevalence of dyslipidemia among HIV patients ranges from 25% (elevated LDL-c) to 61% (overall dyslipidemia). Among hypertensive patients, the prevalence of dyslipidemia varies from 39% (elevated LDL-c) to 70% (overall dyslipidemia). For diabetic patients, the highest and lowest prevalence of dyslipidemia is 55% and 32%, respectively.

## Small study effects

To assess small-study effects, we initially visually inspected the funnel plots, which revealed asymmetry for elevated TC and LDL-C. Subsequently, Doi plots and LFK indices were computed, alongside with Egger's regression p-values. Similarly, Egger's regression test confirmed the presence of asymmetry among the included studies for elevated TC (P < 0.001) and LDL-c (P < 0.001). The LFK index also indicated a major asymmetry for elevated TC (LFK index = 2.47) and LDL-c (LFK index = 2.32). The LFK index for overall dyslipidemia (LFK index= -1.65) showed a minor asymmetry. However, no asymmetry was identified for elevated TG (LFK index= 0.53) and low HDL-c (LFK index = 0.43) (S3 Fig and S4 Fig).

## Discussion

This is the first systematic review and meta-analysis study on dyslipidemia, its components and associated factors among adults with coexisting chronic disease in Ethiopia. Our result showed 64% of the patients have at least one lipid abnormality, and pooled prevalence of elevated TC, elevated TG, elevated LDL-c, and low HDL-c were 32%, 43%, 30%, and 48%, respectively. In this study, the most prominent form of dyslipidemia component was HDL-c, an independent risk factor for CVD. The highest prevalence of low HDL-c phenomenon might reflect a sedentary lifestyle accompanied by high consumption of refined and unhealthy foods that are rich in saturated fat and trans-fatty acids [86]. Having high wrist circumference, older age, being overweight or obese, residing in urban, being alcohol consumers, being smokers, and being physically inactive were significant risk factors for overall dyslipidemia. Additionally, the geographical stratification of results revealed that regional administrations like Sidama, Amhara, and Tigray had a higher prevalence of dyslipidemia compared to the rest of the regional administrations in Ethiopia. Furthermore, our analysis also showed that patients with hypertension and diabetes had a higher prevalence of dyslipidemia. In the present review, the prevalence of overall dyslipidemia (64%) is significantly higher than those reported in china (42.1%) [87], Sub-Saharan Africa (38.38%) [88], Italy (17.5%) [89] and Africa (16.5%) [90]. Our finding also showed a substantially higher prevalence of overall dyslipidemia across different parts of Africa: East Africa (60.8%), Southern (53.1%), west (46.7%), and Central Africa (21%) [90]. Likewise, the prevalence of dyslipidemia components are markedly higher than those observed in Africa [5], China [91], and Malaysia [6]. However, the Malaysian study reported a higher prevalence of elevated LDL-c (73%) and TC (52%). Moreover, our study also revealed a higher prevalence of overall dyslipidemia and its components among patients with hypertension, HIV, and diabetes. Similarly, the prevalence of dyslipidemia components in hypertensive and HIV patients were notably higher compared to the findings of a meta-analysis on dyslipidemia among adults in Africa, which found a prevalence of dyslipidemia components among adult hypertensive patients (elevated TC 38%, elevated TG 22.2%, elevated LDL-c 32.7%, low HDL-c 39.4%) and (elevated TC 24.2%, elevated TG 22.7%, elevated LDL-c 20.8%) among HIV patients [5].

Consistent with this, we observed a higher prevalence of elevated triglycerides (55%) among diabetic patients compared to finding from a study conducted in Ethiopia on diabetic individuals [92]. Our findings, however, were lower than a finding from a study in Ethiopia, which reported TC, LDL-c, and HDL-c levels of 34.08%, 41.13%, and 44.36%, respectively [92]. Discrepancy in prevalence estimates across studies can be attributed to the differences in diagnostic cut-offs, dyslipidemia components assessed, study settings, populations, period, and differences in lifestyle.

Another important aspect of this study is that 19 variables were extracted from 11 studies to determine factors associated with dyslipidemia. Our data demonstrated that 11 of the variables (age, residence, physical activity, current alcohol consumption, history of alcohol consumption, smoking, history of smoking, being hypertensive, high HC, high WC, and

BMI) were significantly associated with overall dyslipidemia. This was in agreement with the findings of a study on determinants of dyslipidemia in Africa, where patients with hypertension, BMI, and high WC were significantly associated with dyslipidemia [90]. No association was found between sex, educational status, marital status, chat chewing, helicobacter pylori infection, family history of DM, family history of DM, and duration of DM with overall dyslipidemia. On the contrary, Obsa et al. reported that sex, educational status, and marital status were significantly associated with the odds of having dyslipidemia [90]. As most of risk factors are modifiable, excluding age, it is crucial to deliver behavioral interventions targeting the identified risk factors. It is highly recommended that performing physical activity, reducing alcohol consumption, and quitting smoking will help to prevent and control dyslipidemia [21].

In this meta-analysis, the finding of high overall dyslipidemia and its components in adults with coexisting chronic disease is hardly wondering given the rapid urbanization, dietary and behavioral changes, and widespread adoption of unhealthy lifestyles occurring across Ethiopia [93, 94]. Literacy status and lack of awareness of common cardiovascular risk factors may also play an important roles and should be considered in prevention and treatment approaches of dyslipidemia [95]. As dyslipidemia is a well-established risk factor for cardiovascular diseases [91], these results highlight the urgent need for appropriate interventions, both clinical and non-clinical. If left uncontrolled, it could result in serious health problems [96]. In this context, dyslipidemia not only fuels the epidemic of CVD but also promotes the progressiveness of adverse outcomes. Access to lipid-lowering drugs remains limited for substantial portions of the world's population, especially in low-income countries such as Ethiopia [2]. Our findings underscore the critical need to address dyslipidemia among adults with coexisting chronic diseases in Ethiopia. This highlights the urgent need for an integrated approach to the prevention, detection, and management of dyslipidemia in the country. This finding is believed to inform clinicians, public health programmers, and policy makers about dyslipidemia and its components burden in adults with coexisting chronic disease in Ethiopia.

This novel study has several strengths including a large sample size and the assessment of dyslipidemia and its components across various patient groups. This approach is particularly valuable as it goes beyond limiting the analysis to a single patient subgroup, such as those with diabetes or hypertension. Despite the important evidence generated by this meta-analysis, our study should be interpreted in light of some limitations. The considerable heterogeneity observed among the included studies was not fully explained through sensitivity or subgroup analyses based on the cut-off used to define dyslipidemia, region and types of coexisted chronic diseases. The heterogeneity between studies is inevitable [97] and could arise from difference in study settings, sample sizes, and the method used to measure dyslipidemia. Another potential limitation is the generalizability of the study findings may be limited, as the included participants were not representative of the national population.

## Conclusions

Our study highlights an alarmingly higher prevalence of lipid abnormalities in adult patients with coexisting chronic diseases in Ethiopia. Consequently, dyslipidemia and its components represent a significant public health challenge in the country. To address this issue effectively, it is crucial to implement measures that improve the prevention, early detection, treatment, and management of dyslipidemia in adults living in Ethiopia. These actions will help mitigate the health and socioeconomic consequences associated with this condition.

## Supporting information

**S1 Table.  PRISMA 2020 checklist.**
(DOCX)

**S2 Table.  Search strategy used for the electronic databases.**
(DOCX)

**S3 Table. Final Joanna Briggs Institute's (JBI's) critical appraisal checklist of included studies.**
(PDF)

**S1 Fig. Sub-group analysis by region.** Forest plots of pooled prevalence of overall dyslipidemia and dyslipidemia components among patients in Ethiopia by region. **A,** Forest plot displaying the pooled prevalence of overall dyslipidemia by region. **B,** Forest plot displaying the pooled prevalence of TC by region. **C,** Forest plot displaying the pooled prevalence of TG by region. **D,** Forest plot displaying the pooled prevalence of LDL-c by region. **E,** Forest plot displaying the pooled prevalence of HDL-c by region.
(TIF)

**S2 Fig. Sub-group analysis by HDL-c cut-off and study population.** Forest plots of pooled prevalence of overall dyslipidemia and dyslipidemia components among patients in Ethiopia by sub-groups. **A,** Forest plot displaying the pooled prevalence of overall dyslipidemia by HDL-c cut-off. **B,** Forest plot displaying the pooled prevalence of TC by study population. **C,** Forest plot displaying the pooled prevalence of TG by study population. **D,** Forest plot displaying the pooled prevalence of LDL-c by study population. **E,** Forest plot displaying the pooled prevalence of HDL-c by study population.
(TIF)

**S3 Fig. Small study effect (Funnel plot).** Small study effect assessed by using funnel plot for overall dyslipidemia and dyslipidemia components among patients in Ethiopia, 2023. **A,** Small study effect assessed by using funnel plot for overall dyslipidemia. **B,** Small study effect assessed by using funnel plot for TC. **C,** Small study effect assessed by using funnel plot for TG. **D,** Small study effect assessed by using funnel plot for LDL-c. **E,** Small study effect assessed by using funnel plot for HDL-c.
(TIF)

**S4 Fig. Small study effect (Doi plot and LFK index).** Small study effect assessed by using the LFK index for overall dyslipidemia and dyslipidemia components among patients in Ethiopia, 2023. **A,** Small study effect assessed by using the LFK index for overall dyslipidemia. **B,** Small study effect assessed by using the LFK index for TC. **C,** Small study effect assessed by using the LFK index for TG. **D,** Small study effect assessed by using the LFK index for LDL-c. **E,** Small study effect assessed by using the LFK index for HDL-c.
(TIF)

**S1 File. Raw data on dyslipidemia prevalence and risk factors in Ethiopian adults with chronic diseases.**
(XLSX)

**S2 File. All studies identified in the literature search.**
(XLSX)

## Author contributions

**Conceptualization:** Tadesse Mekonen Zeleke, Fitsum Endale Liben, Tinsae Shemelise Tesfaye, Abiyu Legesse Munea.

**Data curation:** Tadesse Mekonen Zeleke, Fitsum Endale Liben, Tinsae Shemelise Tesfaye.

**Formal analysis:** Tadesse Mekonen Zeleke, Fitsum Endale Liben, Tinsae Shemelise Tesfaye, Fekede Asefa Kumsa.

**Investigation:** Tadesse Mekonen Zeleke.

**Methodology:** Tadesse Mekonen Zeleke, Fitsum Endale Liben, Tinsae Shemelise Tesfaye, Abiyu Legesse Munea, Fekede Asefa Kumsa.

**Project administration:** Tadesse Mekonen Zeleke, Fekede Asefa Kumsa.

**Resources:** Tadesse Mekonen Zeleke.

**Software:** Tadesse Mekonen Zeleke, Tinsae Shemelise Tesfaye, Fekede Asefa Kumsa.

**Supervision:** Tadesse Mekonen Zeleke, Fitsum Endale Liben, Abiyu Legesse Munea, Fekede Asefa Kumsa.

**Validation:** Tadesse Mekonen Zeleke, Fitsum Endale Liben.

**Visualization:** Tadesse Mekonen Zeleke, Fekede Asefa Kumsa.

**Writing – original draft:** Tadesse Mekonen Zeleke, Tinsae Shemelise Tesfaye.

**Writing – review & editing:** Tadesse Mekonen Zeleke, Abiyu Legesse Munea, Fekede Asefa Kumsa.

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
