## [Decision Letter · Decision Letter 0]

7 Nov 2024

PONE-D-24-01697Dyslipidemia and associated factors among adults with coexisting chronic disease in Ethiopia: a systematic review and meta-analysisPLOS ONE

Dear Dr. Zeleke,

Thank you for submitting your manuscript to PLOS ONE. After careful consideration, we feel that it has merit but does not fully meet PLOS ONE’s publication criteria as it currently stands. Therefore, we invite you to submit a revised version of the manuscript that addresses the points raised during the review process.

Please carefully address all the issues raised by reviewer 2.

We look forward to receiving your revised manuscript.

Kind regards,

Paolo Magni

Academic Editor

PLOS ONE

Journal Requirements:

2. As required by our policy on Data Availability, please ensure your manuscript or supplementary information includes the following: 

Additional Editor Comments:

The paper is interesting, but requires an extensive revision prior to be considered for potential publication. Please address all issues raised by reviewer 2.

Reviewers' comments:

Reviewer's Responses to Questions

**Comments to the Author**

1. Is the manuscript technically sound, and do the data support the conclusions?

Reviewer #1: Yes

Reviewer #2: Yes

2. Has the statistical analysis been performed appropriately and rigorously? 

Reviewer #1: Yes

Reviewer #2: Yes

3. Have the authors made all data underlying the findings in their manuscript fully available?

Reviewer #1: Yes

Reviewer #2: Yes

4. Is the manuscript presented in an intelligible fashion and written in standard English?

Reviewer #1: Yes

Reviewer #2: Yes

5. Review Comments to the Author

Reviewer #1: This metanalysis studies already known risk factors. Nothing new is added to the already existing knowledge. Most of the studies included are in specific populations. Hence the results are not generalizable .

Reviewer #2: General:

This research is highly relevant and timely, given the rising burden of dyslipidemia as a modifiable risk factor for cardiovascular diseases, particularly in low-resource settings like Ethiopia. Understanding the prevalence and associated risk factors is crucial for developing targeted public health interventions. Your study contributes valuable insights into this pressing health issue.

Title:

The title effectively conveys the study's focus on dyslipidemia among adults with chronic diseases in Ethiopia. However, consider specifying "prevalence" or "risk factors" to enhance clarity.

Abstract:

The abstract is well-structured and provides a clear overview of the study’s objectives, methods, results, and conclusions. However, breaking up some of the longer sentences could improve readability.

Introduction:

The introduction offers a comprehensive overview of dyslipidemia and its implications. Streamlining certain sentences could enhance clarity and flow. For example, transitioning between the global context and the specific situation in Ethiopia could be smoother.

The statistics on dyslipidemia prevalence in Africa and Ethiopia are compelling. It would be beneficial to briefly mention healthcare challenges in Ethiopia—such as access to care and lifestyle changes—that contribute to the rising prevalence.

Linking the statistics on cardiovascular disease (CVD) more directly to the Ethiopian context would strengthen the urgency of your argument. Citing local data on CVD mortality could be particularly impactful.

While the references are relevant, ensure that they are current and accessible. Additionally, highlighting the significance of your findings for public health policy or clinical practice at the end of the introduction would provide a stronger rationale for the study.

Recommendations:

- Streamline sentences for clarity and flow.

- Provide context regarding healthcare challenges in Ethiopia.

- Directly link global statistics to the Ethiopian situation.

- Ensure references are up to date.

- Highlight the significance of your findings for public health or clinical practice more explicitly.

Methods

The methods section is well-organized and adheres to PRISMA guidelines, which is commendable. However, simplifying some lengthy sentences could improve clarity.

The inclusion and exclusion criteria are well-defined, but it would be helpful to briefly justify why certain populations (e.g., children and adolescents) are excluded, especially regarding dyslipidemia prevalence.

While the definition of dyslipidemia is clear, it would be useful to highlight the varying cut-off values across studies as a potential source of variability in prevalence estimates. Discussing how this variability might impact the results could add depth.

The study selection and data extraction processes are well-articulated. Mentioning specific software used for data extraction, in addition to Excel, would enhance the methodological rigor.

Results and Discussion:

The prevalence rates are striking and provide a solid overview. To enhance clarity, consider stating, "Our prevalence rates are significantly higher than those reported in China and Sub-Saharan Africa."

The discussion on lifestyle factors contributing to low HDL-c is insightful. Expanding on how these choices are influenced by socio-economic and cultural factors in Ethiopia could strengthen your argument.

The comparisons with other meta-analyses and studies are valuable. Highlighting potential reasons for discrepancies in prevalence findings—such as differences in study populations or methodologies—would provide important context.

While the identification of significant risk factors is well-articulated, mentioning the public health implications of these modifiable risk factors—such as potential interventions—earlier in the discussion could enhance the practical relevance of your findings.

Limitations:

Acknowledging the considerable heterogeneity among studies is essential for transparency. Briefly discussing potential sources of this heterogeneity—such as differences in study design or diagnostic criteria—could give readers a clearer understanding of the limitations.

While it's good to note that subgroup and sensitivity analyses were performed, elaborating on what aspects of heterogeneity remain unexplained could provide more context for those interested in the methodological rigor of your analysis.

Overall, this research holds significant potential to inform public health strategies and interventions in Ethiopia. Thank you for your important work in this area!

6. PLOS authors have the option to publish the peer review history of their article (what does this mean? ). If published, this will include your full peer review and any attached files.

**Do you want your identity to be public for this peer review?** For information about this choice, including consent withdrawal, please see our Privacy Policy .

Reviewer #1: **Yes: ** Dr. Archith Boloor

Reviewer #2: **Yes: ** Natan Mulubrhan Alemseged

---

## [Author Response · Author response to Decision Letter 1]

13 Jan 2025

Feedback of reviewers

Reviewer #1 (Comments to the Author):

Thank you for your comments and suggestions. We truly appreciate your time in reviewing our paper.

Reviewer #1: This metanalysis studies already known risk factors. Nothing new is added to the already existing knowledge. Most of the studies included are in specific populations. Hence the results are not generalizable.

Response: this review is the first of its kind to provide a comprehensive analysis of dyslipidemia prevalence and its components (elevated TC, TG, LDL-c, and low HDL-c) among adults with coexisting chronic diseases in Ethiopia. Notably, we present pooled prevalence estimates across various chronic diseases and geographic regions, offering new insights into the distribution and burden of dyslipidemia. However, the generalizability of the study findings may be limited because the regional representation was uneven, with the majority of the studies (43 out of 62) originating from Addis Ababa city (the capital city) and two regional states (Amhara and Oromia). Also, there was lack of studies in certain regions of the country. This limitation may affect the generalizability of our findings, as addressed in our manuscript in lines 339 - 341 “Another potential limitation is the generalizability of the study findings, as the included participants were not representative of the national population.” Despite the limitations, we strongly believe that the findings make a valuable contribution to the existing body of literature.

Reviewer #2: General: This research is highly relevant and timely, given the rising burden of dyslipidemia as a modifiable risk factor for cardiovascular diseases, particularly in low-resource settings like Ethiopia. Understanding the prevalence and associated risk factors is crucial for developing targeted public health interventions. Your study contributes valuable insights into this pressing health issue.

Thank you for your thoughtful comments and suggestions. We sincerely appreciate the time and effort you dedicated to review our paper.

Title: The title effectively conveys the study's focus on dyslipidemia among adults with chronic diseases in Ethiopia. However, consider specifying "prevalence" or "risk factors" to enhance clarity.

Response: we have revised the title in lines 1-2, which now read “Prevalence and associated factors of dyslipidemia among adults with coexisting chronic disease in Ethiopia: a systematic review and meta-analysis”

Abstract:

The abstract is well-structured and provides a clear overview of the study’s objectives, methods, results, and conclusions. However, breaking up some of the longer sentences could improve readability.

Response: we agree with the reviewer’s point on the importance of breaking up of some of the longer sentences. We have revised the whole abstract accordingly, and for instance lines 33 - 34, now read “Out of 640 initially identified articles, 104 underwent full-text review, and 62 met the inclusion criteria for the final analysis.”

Introduction:

The introduction offers a comprehensive overview of dyslipidemia and its implications. Streamlining certain sentences could enhance clarity and flow. For example, transitioning between the global context and the specific situation in Ethiopia could be smoother.

Response: effort has been made to streamline sentences in the introduction part to enhance flow and clarity, line 62-69 now read “CVD is a chronic non-communicable disease (NCD) and one of the leading causes of global mortality. According to the American Heart Association, the annual number of CVD-related deaths was 19 million in 2020 and is estimated to reach 22.2 million in 2030 (12, 13). Over the past four decades, there has been a significant surge in the prevalence of cardiovascular diseases (CVD) in Africa. And it is anticipated to overtake infectious diseases as the leading cause of death by 2030 (14). Ethiopia has made inadequate progress in reducing cardiovascular and other non-communicable diseases, with CVDs are one of the leading causes of premature mortality and death (15, 16).”

The statistics on dyslipidemia prevalence in Africa and Ethiopia are compelling. It would be beneficial to briefly mention healthcare challenges in Ethiopia—such as access to care and lifestyle changes—that contribute to the rising prevalence.

Response: healthcare challenges were considered as one of contributing factors for increased prevalence of dyslipidemia in Ethiopia. Please see lines 69 - 73: “Access to public healthcare facilities and availability of health services in Ethiopia are generally limited. Furthermore, there are substantial disparities in accessibility and inequalities in the distribution of healthcare resources (17-19). These healthcare challenges might also contribute to the rising prevalence of dyslipidemia and present significant barriers to its prevention and management.”

Linking the statistics on cardiovascular disease (CVD) more directly to the Ethiopian context would strengthen the urgency of your argument. Citing local data on CVD mortality could be particularly impactful.

Response: a sentence reflecting CVDs mortality in Ethiopia is added and line 67 - 69 read: “Ethiopia has made inadequate progress in reducing cardiovascular and other non-communicable diseases, with CVDs are one of the leading causes of premature mortality and death (15, 16).”

While the references are relevant, ensure that they are current and accessible.

Response: we have reviewed all the references and ensure they are up-to-date and accessible.

Additionally, highlighting the significance of your findings for public health policy or clinical practice at the end of the introduction would provide a stronger rationale for the study.

Response: we have added the significance of the study to the last paragraph of the introduction (line 74 - 80) which now reads “This review will enable researchers to evaluate the burden of dyslipidemia, where existing evidence is often hospital-based, fragmented and inconsistent. Understanding the prevalence and factors associated with dyslipidemia can also provide valuable insights for researchers and policymakers, in designing targeted interventions and strategies for effective disease management and the prevention of further complications. Therefore, this study aimed to systematically review and pool the prevalence of dyslipidemia, its components, and associated factors among adults with coexisting chronic disease in Ethiopia.”

Recommendations:

- Streamline sentences for clarity and flow.

- Provide context regarding healthcare challenges in Ethiopia.

- Directly link global statistics to the Ethiopian situation.

- Ensure references are up to date.

- Highlight the significance of your findings for public health or clinical practice more explicitly.

Response: we have already addressed the points raised in the recommendation in the responses above.

Methods

The methods section is well-organized and adheres to PRISMA guidelines, which is commendable.

However, simplifying some lengthy sentences could improve clarity.

Response: we appreciate the reviewer’s point on the importance of simplifying lengthy sentences. The whole method section have been edited accordingly: For instance line 124 - 125 read “To ensure data quality and reliability, two independent reviewers (TMZ and FEL) conducted the extraction process.”

The inclusion and exclusion criteria are well-defined, but it would be helpful to briefly justify why certain populations (e.g., children and adolescents) are excluded, especially regarding dyslipidemia prevalence.

Response: exclusion based on age was performed because of not having eligible studies on dyslipidemia in children and adolescents with coexisting chronic disease in Ethiopia. Now, we have tried to include justification and line 98 - 101 reads “We included original studies reporting the prevalence and factors associated with overall dyslipidemia, elevated TC, LDL-c, TG, or low HDL-c among adult patients (aged ≥ 18 years) in Ethiopia, as no eligible studies on dyslipidemia in children and adolescents with coexisting chronic illnesses were identified.”

While the definition of dyslipidemia is clear, it would be useful to highlight the varying cut-off values across studies as a potential source of variability in prevalence estimates. Discussing how this variability might impact the results could add depth.

Response: we agree that various studies used various diagnostic cutoff values to define dyslipidemia. However, this study considered those articles which define dyslipidemia based on the National Cholesterol Education Program Adult Treatment Panel III cut-off points. Besides, we further considered the two most frequently used HDL cut-off values (HDL < 40 mg/dl, and HDL < 40 mg/dl for males and < 50 mg/dl for female). Hence, based on our sub group analyses by HDL cut-off values, the pooled prevalence of overall dyslipidemia was high, regardless of the HDL-c cut-off used. The prevalence was 68% (0.56 – 0.81, I2 = 98.39) and 60% (0.45 – 0.75, I2 = 98.80) for studies used HDL < 40 mg/dl, and HDL < 40 mg/dl for males and < 50 mg/dl for female, respectively (S4 Appendix (A)). Based on this finding, it is difficult to consider dyslipidemia cut-off values as potential source of heterogeneity (I2 = 98.39 for HDL < 40 mg/dl and I2 = 98.80 for HDL < 40 mg/dl for males and < 50 mg/dl for female). Furthermore, the confidence interval for the two prevalence estimate overlaps and we assume that the pooled prevalence of overall dyslipidemia isn’t varied significantly by utilizing the two HDL cut-off values.

The study selection and data extraction processes are well-articulated. Mentioning specific software used for data extraction, in addition to Excel, would enhance the methodological rigor.

Response: we utilized covidence, a web-based software, for study selection and duplicate removal. This has already been detailed in the Methods section, as indicated in the corresponding line 113 - 115 “Studies were identified using search terms and database filters, then imported them into covidence, a systematic review software (Melbourne, Australia, Veritas Health Innovation) to remove duplicates and

conduct screening.

Results and Discussion:

The prevalence rates are striking and provide a solid overview. To enhance clarity, consider stating, "Our prevalence rates are significantly higher than those reported in China and Sub-Saharan Africa."

Response: we have revised part of the discussion accordingly and for instance, sentence in lines 382 - 284, which now read “In the present review, the prevalence of overall dyslipidemia (64%) is significantly higher than those reported in china (42.1%) (87), Sub-Saharan Africa (38.38%) (88), Italy (17.5%) (89) and Africa (16.5%) (90).”

The discussion on lifestyle factors contributing to low HDL-c is insightful. Expanding on how these choices are influenced by socio-economic and cultural factors in Ethiopia could strengthen your argument.

Response: we totally agree with the comment, yet we have tried to discuss about socio-economic and cultural factors in the final paragraph of discussion section (Line 316 - 321) “In this meta-analysis, the finding of high overall dyslipidemia and its components in adults with coexisting chronic disease is hardly wondering given the rapid urbanization, dietary and behavioral changes, and widespread adoption of unhealthy lifestyles occurring across Ethiopia (93, 94). Literacy status and lack of awareness of common cardiovascular risk factors may also play an important roles and should be considered in prevention and treatment approaches of dyslipidemia (95).”

The comparisons with other meta-analyses and studies are valuable. Highlighting potential reasons for discrepancies in prevalence findings—such as differences in study populations or methodologies—would provide important context.

Response: we have included the potential reasons for discrepancy in line 299 – 301, now read “Discrepancy in prevalence estimates across studies can be attributed to the differences in diagnostic cut-offs, dyslipidemia components assessed, study settings, populations, period, and differences in lifestyle.”

While the identification of significant risk factors is well-articulated, mentioning the public health implications of these modifiable risk factors—such as potential interventions—earlier in the discussion could enhance the practical relevance of your findings.

Response: we accept the feedback and revised line 312 - 315: “As most of risk factors are modifiable, excluding age, it is crucial to deliver behavioral interventions targeting the identified risk factors. It is highly recommended that performing physical activity, reducing alcohol consumption, and quitting smoking will help to prevent and control dyslipidemia (21).”

Limitations:

Acknowledging the considerable heterogeneity among studies is essential for transparency. Briefly discussing potential sources of this heterogeneity—such as differences in study design or diagnostic criteria—could give readers a clearer understanding of the limitations.

While it's good to note that subgroup and sensitivity analyses were performed, elaborating on what aspects of heterogeneity remain unexplained could provide more context for those interested in the methodological rigor of your analysis.

Overall, this research holds significant potential to inform public health strategies and interventions in Ethiopia. Thank you for your important work in this area!

Response: we have acknowledged heterogeneity among studies in lines 336 - 340: “The considerable heterogeneity observed among the included studies was not fully explained through sensitivity or subgroup analyses based on the cut-off used to define dyslipidemia, region and types of coexisted chronic diseases. The heterogeneity between studies is inevitable (97) and could arise from difference in study settings, sample sizes, and the method used to measure dyslipidemia.”

---

## [Decision Letter · Decision Letter 1]

14 Feb 2025

Prevalence and associated factors of dyslipidemia among adults with coexisting chronic disease in Ethiopia: a systematic review and meta-analysis

PONE-D-24-01697R1

Dear Dr. Tadesse Mekonen Zeleke,

We’re pleased to inform you that your manuscript has been judged scientifically suitable for publication and will be formally accepted for publication once it meets all outstanding technical requirements.

Kind regards,

Paolo Magni

Academic Editor

PLOS ONE

Additional Editor Comments (optional):

Reviewers' comments:

Reviewer's Responses to Questions

**Comments to the Author**

1. If the authors have adequately addressed your comments raised in a previous round of review and you feel that this manuscript is now acceptable for publication, you may indicate that here to bypass the “Comments to the Author” section, enter your conflict of interest statement in the “Confidential to Editor” section, and submit your "Accept" recommendation.

Reviewer #1: All comments have been addressed

2. Is the manuscript technically sound, and do the data support the conclusions?

Reviewer #1: Yes

3. Has the statistical analysis been performed appropriately and rigorously? 

Reviewer #1: Yes

4. Have the authors made all data underlying the findings in their manuscript fully available?

Reviewer #1: Yes

5. Is the manuscript presented in an intelligible fashion and written in standard English?

Reviewer #1: Yes

6. Review Comments to the Author

Reviewer #1: (No Response)

7. PLOS authors have the option to publish the peer review history of their article (what does this mean? ). If published, this will include your full peer review and any attached files.

**Do you want your identity to be public for this peer review?** For information about this choice, including consent withdrawal, please see our Privacy Policy .

Reviewer #1: **Yes: ** Dr. Archith Boloor

---

## [Editor Report · Acceptance letter]

PONE-D-24-01697R1

PLOS ONE

Dear Dr. Zeleke,

I'm pleased to inform you that your manuscript has been deemed suitable for publication in PLOS ONE. Congratulations! Your manuscript is now being handed over to our production team.

Kind regards,

on behalf of

Prof. Paolo Magni

Academic Editor

PLOS ONE